# DYNAMIC ADAPTER MERGING FOR CONTINUAL VIDEO QUESTION-ANSWERING LEARNING

## ABSTRACT

We present a parameter-efficient method for continual video question-answering (VidQA) learning. Our method, named DAM, uses **D**ynamic **A**dapter **M**erging to address the issues of (i) catastrophic forgetting, (ii) the costly retraining of large VidQA models on continually shifting distribution of training data, and (iii) handling inputs from an unknown domain during test-time inference. Given a set of different VidQA datasets, we sequentially train domain-specific adapters for each VidQA dataset while freezing the parameters of a large pretrained video-language backbone. During inference, given a video-question sample from an unknown domain, our method first uses a non-parametric video-language router function to compute a probability for each domain-specific adapter, reflecting how relevant that adapter is to the current video-question input instance. Afterward, to exploit beneficial cross-domain cues and reduce the impact of potentially incorrect router predictions, we dynamically merge the parameters of several highest-scoring adapters for the final VidQA prediction. Despite the simplicity of our approach, we demonstrate that it works well on continually streaming VidQA datasets across 6 different domains. In particular, our model outperforms prior prompt-based continual learning approaches by 9.1% while exhibiting 1.9% less forgetting. The code and pretrained models will be publicly released.

## 1 INTRODUCTION

In recent years, Video Question-Answering (VidQA) has advanced significantly due to large-scale video-language pretraining datasets (Sharma et al., 2018; Miech et al., 2019; Bain et al., 2021) and the emergence of large video-language models (Yu et al., 2021; Yang et al., 2022; Cheng et al., 2023). Modern VidQA models commonly follow the *pretrain-finetune* paradigm (Cheng et al., 2023; Lei et al., 2021; Li et al., 2020; Miech et al., 2019; Sun et al., 2019). This involves initial pretraining on extensive paired video-language data and subsequent fine-tuning on domain-specific VidQA datasets. However, this approach necessitates managing numerous domain-specific fine-tuned models, incurring substantial complexity and cost, especially for scenarios involving many domains and datasets. Moreover, modern VidQA models often assume static conditions with fixed training and testing datasets. However, real-world applications increasingly demand adaptability to dynamic shifts in training data distribution. For instance, a VidQA model trained only on Instagram videos may struggle when questioned about the recently released "Barbie" movie (Fig. 1). This difficulty arises due to domain disparities (Instagram vs. movies) and the temporal gap, as most VidQA models were trained on data collected before 2023, the year of the "Barbie" movie's release.

To address this issue, one could finetune a VidQA model every time new training data is added. However, this is problematic for two main reasons. Firstly, it often leads to the model forgetting previously learned information, a phenomenon known as *catastrophic forgetting* (McClelland et al., 1995; McCloskey & Cohen, 1989). Secondly, fine-tuning an entire VidQA model, which can contain billions of parameters, for each new dataset incurs substantial computational costs. It's worth noting that these computational challenges are exacerbated in the video domain due to the high-dimensional nature of video data and the resource-intensive design of modern VidQA model architectures (Zellers et al., 2021; Fu et al., 2021; Li et al., 2023c; Wang et al., 2022a).

Motivated by these challenges, we delve into the domain of continual VidQA learning. Our specific focus lies on tackling the rehearsal-free Domain-Incremental Learning (DIL) subproblem of con-

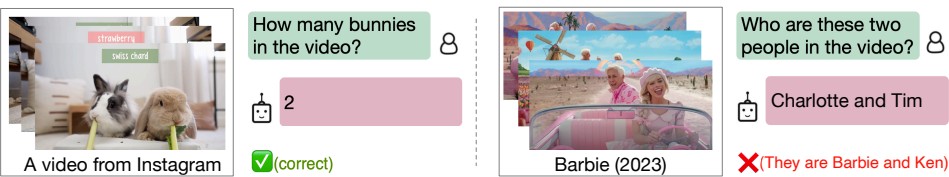

Figure 1: Given a question about an Instagram video, a video question-answering (VidQA) model trained only on Instagram videos will likely answer that question correctly. However, the same VidQA model will fail to answer a question about a video clip from the "Barbie" movie due to (i) disparities in video domains (Instagram vs. movies), and (ii) the fact that the model was trained on videos collected before 2023, predating the release of the "Barbie" movie. This highlights the limitations of most modern VidQA models in adapting to the continually shifting data distributions.

tinual learning (Kirkpatrick et al., 2017; Wang et al., 2023). In DIL, the model must continuously adapt to a sequence of datasets spanning different domains. During inference, given a sample from an unknown domain, the model must discern the most relevant domain and provide a final output, such as answering a video-related question. Recent DIL methods (Wang et al., 2022b;e; Douillard et al., 2022; Smith et al., 2023a) have proposed techniques involving domain-specific prompts and a router for prompt selection during inference. However, these methods exhibit suboptimal performance when the router erroneously selects prompts. Additionally, these prior approaches are primarily tailored for image classification tasks, characterized by relatively minor variations between dataset domains, sizes, and other factors. In stark contrast, VidQA is a more formidable challenge, requiring the model to comprehend both video and language. Furthermore, the DIL VidQA problem is even more challenging due to the disparities between dataset domains, question-answer pair styles, dataset size imbalances, video durations, and more.

To overcome the limitations of previous Domain-Incremental Learning (DIL) approaches and address the challenges of continual VidQA learning, we introduce DAM, a **D**ynamic **A**dapter **M**erging scheme designed for parameter-efficient continual VidQA learning. Our model uses domain-specific adapters and model merging techniques (Wortsman et al., 2022a; Matena & Raffel, 2022) to tackle several critical issues: (i) mitigating catastrophic forgetting, (ii) reducing the substantial retraining cost associated with modern VidQA models as training data evolves, and (iii) handling the challenge of unknown input domains during test-time inference. Given a sequence of VidQA datasets from different domains, we begin by training a set of domain-specific adapters for each VidQA dataset while freezing the parameters of a pretrained video-language backbone (e.g., CLIP (Radford et al., 2021) and DeBERTa (He et al., 2020)). During inference, we employ a non-parametric video-language router to estimate probabilities for each domain-specific adapter. These probabilities reflect the relevance of each adapter to that particular video-question input instance. Subsequently, we utilize these adapter probabilities to select the most pertinent domain-specific adapters for each video-question instance from an unknown domain. Our experiments reveal the inherent challenges in domain prediction, where the router frequently generates inaccurate domain predictions, resulting in suboptimal VidQA performance. To address this issue of potentially erroneous router predictions, we introduce a dynamic parameter merging approach. Instead of relying on a single set of domain-specific adapters, we dynamically merge the parameters of multiple sets of adapters with the highest scores for the final VidQA prediction. This dynamic merging scheme not only mitigates the impact of inaccurate router predictions but also facilitates the sharing of valuable VidQA cues across diverse domains, thereby enhancing VidQA performance (refer to Sec. 4.4 for detailed experimentation).

In summary, our contributions are four-fold. Firstly, we are the first to explore domain-incremental VidQA learning, particularly on large-scale models with billions of parameters. Secondly, we propose a novel technique, Dynamic Adapter Merging, which innovatively generates a personalized expert model for each testing sample with minimal overhead. We also performed in-depth analyses detailing how and when model merging can enhance the effectiveness of the router-based technique in the continual learning domain. Thirdly, compared to prior DIL methods, our proposed DAM achieves **9.1%** better results on sequentially-introduced VidQA datasets from 6 different domains while exhibiting **1.9%** less forgetting. Lastly, our method's simplicity and adaptability make it easy to integrate into other tasks (e.g. image question-answering.) and model merging community To enable the community to develop models for this emerging research area of domain-incremental VidQA learning, we will release our code and pretrained models.

## 2 RELATED WORK

**Video Question Answering (VidQA)** represents a fundamental task in video-language understanding, aiming to answer natural language questions based on given videos. Most commonly used methods (Yang et al., 2022; Yu et al., 2023; Xiao et al., 2022; Cheng et al., 2023; Lei et al., 2021; Li et al., 2020; Miech et al., 2019; Sun et al., 2019) construct video-language models (VLMs) with transformer architecture (Xiao et al., 2022; Lei et al., 2021; Cheng et al., 2023) and large pre-trained language models (Yang et al., 2022; Yu et al., 2023). FrozenBiLM (Yang et al., 2022) handles the multimodal input using a pretrained bidirectional language model and casts VidQA as a masked language modeling problem. SeViLA (Yu et al., 2023) is built upon a large image-language model, BLIP-2 (Li et al., 2023b), and extends it to accommodate video input for VidQA. To our knowledge, our work is the very first exploration of the domain-incremental VidQA learning problem.

**Continual Learning (CL)** focuses on developing frameworks that can continually learn new information from streaming training datasets. This is a fundamental challenge for many deep learning methods due to *catastrophic forgetting* (McClelland et al., 1995). Continual learning methods can be categorized into regularization-based approaches (Kirkpatrick et al., 2017; Li & Hoiem, 2017), replay-based approaches (Cha et al., 2021a; Riemer et al., 2018), optimization-based approaches (Lopez-Paz & Ranzato, 2017; Chaudhry et al., 2018) and representation-based approaches (Gao et al., 2023; Foret et al., 2020; Ermis et al., 2022; Douillard et al., 2022). Several recent CL approaches use pre-trained models for the vision-language domain, including CLiMB (Srinivasan et al., 2022) for task-incremental learning, VQACL (Zhang et al., 2023) and CL-CrossVQA (Zhang et al., 2022) for rehearsal-based Domain-Incremental Learning (DIL). Rehearsal-based methods require storing some data of previously trained domains, which may not be realistic as the data may be private or limited by intellectual property. In contrast, rehearsal-free CL approaches (Li & Hoiem, 2017; Smith et al., 2023b; 2021) are learned without storing training data of previously learned domains. Several recent prompt-based methods in this area such as L2P (Wang et al., 2022e), DualPrompt (Wang et al., 2022d), S-Prompts (Wang et al., 2022b) and CODA-Prompt (Smith et al., 2023a) employed visual prompts (Liu et al., 2023) prepended to a pre-trained transformer and extended prompt-based learning for continual learning scenarios. Compared to these prior image-level approaches, we focus on rehearsal-free DIL for VidQA, which is more challenging as it typically includes more diverse datasets from different domains. Furthermore, unlike prior prompt-based DIL methods, we use dynamic model merging to alleviate the issues of inaccurate router predictions and enable cross-domain knowledge sharing.

**Model Merging** aims to merge multiple domain models into a single model that can be used for inference on these domains. For instance, the work in (Wortsman et al., 2022b; Ilharco et al., 2022b) computes the merged weights as an element-wise arithmetic mean of the weights of all domain models. Subsequently, several methods proposed to improve the performance of the model merging using techniques such as Fisher Merging (Matena & Raffel, 2022), RegMean (Jin et al., 2022), Git Re-Basin (Ainsworth et al., 2022), Task Arithmetic (Ilharco et al., 2022a) and TIES-Merging (Yadav et al., 2023). Model merging has been applied to many scenarios, including federated learning (McMahan et al., 2017), improving out-of-domain generalization (Cha et al., 2021b), and improving performance on a single target task (Gupta et al., 2020; Wortsman et al., 2022a). Recently, (Guerrero-Peña et al., 2022) proposes a Sinkhorn re-basin network for replay-based class incremental continual learning but only experiments with small models (e.g., ResNet18 (He et al., 2016)) on small datasets (e.g., CIFAR-100 (Krizhevsky et al., 2009)). In comparison, we adapt model merging techniques to rehearsal-free domain-incremental VidQA learning on large-scale models.

## 3 DYNAMIC ADAPTER MERGING

We focus on rehearsal-free domain incremental learning (DIL) (Wang et al., 2022b;e), where the model is sequentially trained on data from $S$ distinct domains and is then required to generalize to all $S$ domains without forgetting previously acquired knowledge. Formally, let $\mathbb{D}_s = \{x_i^s, y_i^s\}_{i=1}^{N_s}$ represent the dataset for the current domain $s$, where $x_i^s$, $y_i^s$, and $N_s$ denote the input, target, and number of samples, respectively. During the training on domain $s$, the model can only access the data from this domain (i.e., no samples from previously encountered domains can be stored in the memory, as opposed to replay-based approaches). During inference, the model predicts a test sample $x_j$ without prior knowledge of which of the $S$ domains the test sample belongs to.

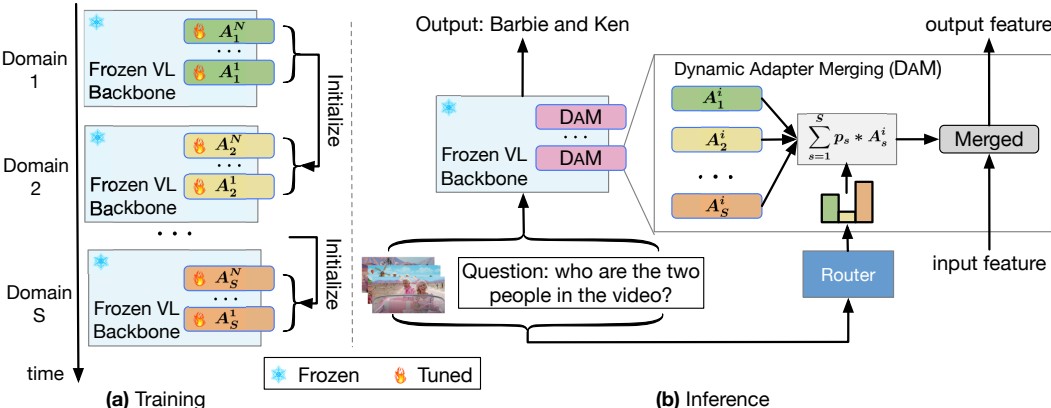

Figure 2: An overview of our Dynamic Adapter Merging (DAM) framework. **(a)** To train our model in a domain-incremental continual learning setting, for each domain $s$, we first inject $N$ domain-specific adapters $\{A_s^1, A_s^2, ... A_s^N\}$ into the frozen video-language backbone. We then sequentially train each domain-specific adapter on the data of its corresponding domain. After each sequential round of training, the weights of subsequent adapter layers are initialized with the weights from the domain-specific adapters that was trained last. **(b)** During inference, given an input video and a text question, we use a non-parametric router function to predict the probability of each adapter being relevant to that particular input instance. Afterward, we dynamically merge multiple domain-specific adapters in parameter space to reduce the impact of incorrect router predictions and leverage cross-domain VidQA cues. Finally, the merged adapter is used to make the final VidQA predictions.

Our proposed **D**ynamic **A**dapter **M**erging framework (DAM) consists of four main components: (i) a frozen pretrained video-language backbone, (ii) continually learned domain-specific adapters, (iii) a non-parametric video-language router that predicts probabilities for selecting the most relevant adapters for a given test-time VidQA input instance, and (iv) a soft parameter-wise adapter merging scheme. At a high level, given a frozen pretrained video-language backbone, for each domain $s$, we first inject $N$ domain-specific adapters into the frozen network. We then sequentially train each domain-specific adapter on its corresponding domain while freezing the parameters of a pretrained video-language backbone. Afterward, during inference, we use a non-parametric router to compute probabilities indicating how relevant each adapter is to a given VidQA input instance. Lastly, we dynamically merge all domain-specific adapters from all domains according to the router-predicted probabilities and use the merged adapter to make a final VidQA prediction for that input instance. In Fig. 2, we present a detailed overview of our approach.

## 3.1 CONTINUALLY LEARNED DOMAIN-SPECIFIC ADAPTERS

Given a VidQA model with a frozen video-language backbone and $S$ continually streaming domains, we incorporate a series of continually learned domain-specific adapters for each domain $s$ as shown in Fig. 2a. Specifically, for each domain $s$, we insert domain-specific adapters after the Self-Attention and Feed-forward Network in each layer of our frozen video-language backbone. We then train such domain-specific adapters continually on datasets from $S$ domains. After each sequential round of training, the weights of the last-trained adapter layers serve as an initialization to the adapter layers for a subsequent domain, which we refer to as a *continual initialization* scheme.

Different from recent DIL methods that aim to keep domain-specific modules (e.g., prompts) independent (Wang et al., 2022b) or even orthogonal (Smith et al., 2023a), our continually learned domain-specific adapters are trained independently (i.e., previously trained adapters will not be updated in subsequent rounds of training) but they also share information via weight inheritance due to the continual weight initialization scheme. There are several benefits of such an approach. First, each set of adapters is trained for a single domain without interfering with the adapters trained on other domains. This prevents catastrophic forgetting as all the past information is preserved, and each adapter can accurately learn representations specialized in its own domain. Second, these domain-specific adapters contain less than $5\%$ of the total parameters of the pretrained model, making the continual learning process scalable and efficient, and also allowing us to integrate our approach with large capacity VidQA models such as FrozenBiLM (Yang et al., 2022). Lastly, due to the continual

initialization scheme (See Fig. 2), each continually learned adapter inherits knowledge from their predecessor adapters (i.e., adapters that were trained before), which is helpful for the subsequent dynamic adapter merging scheme since it leads to a smoother parameter space for continually learned adapters, and a reduction of the interference disagreements (Yadav et al., 2023; Jin et al., 2022).

## 3.2 Non-Parametric Router Function

During inference, we use a non-parametric router to predict the probability of each adapter, estimating how relevant that adapter is to a given video-question input instance from an unknown domain. Specifically, we first calculate the centroid $c_s$ of each domain-specific dataset $\mathbb{D}_s = \{x_i^s, y_i^s\}_{i=1}^{N_s}$ by averaging all multimodal video-language features extracted by a pretrained model $f$:

$$c_s = \frac{1}{N_s} \sum_{i=1}^{N_s} f(x_i^s). \tag{1}$$

Then, during inference, we calculate adapter-specific domain probabilities $p \in \mathbb{R}^S$ by computing the cosine similarity of $f(x)$ and each centroid as:

$$p_s = \frac{\exp(l_s/\tau)}{\sum_{i=1}^{S} \exp(l_i/\tau)}, \tag{2}$$

where $l_s = \cos(f(x), c_s)$ is the cosine similarity between a feature $f(x)$ and a centroid $c_s$, and $\tau$ is the temperature hyper-parameter. Compared to prior DIL methods that use significantly more complex router designs (Smith et al., 2023a; Wang et al., 2022e), our non-parametric router is much simpler yet more effective, as we will show in our experiments. Furthermore, we found that joint end-to-end trainable routers (Smith et al., 2023a) used in prior works often caused optimization stability issues, whereas our simple router did not interfere with the continual learning process.

## 3.3 Merging Domain-specific Adapters

One key challenge in DIL is that the domain identity during test-time inference is unknown. As a result, most recent DIL methods (Wang et al., 2022b;e) require a very accurate router function for selecting which domain a given test sample is most relevant to. However, accurate domain prediction is challenging and typically results in many incorrect predictions that dramatically impact the final DIL performance. As a result, selecting only one domain-specific adapter corresponding to the highest router-predicted probability typically leads to suboptimal VidQA performance, which we demonstrate in our experimental analysis in in Sec. 4.3.

To address this issue, we propose dynamically merging multiple domain-specific adapters for each test-time input instance (Fig. 2b). Our scheme for merging domain-specific adapters is implemented via a simple instance-wise adapter weight merging using soft router-predicted probabilities. Note that all domain-specific adapters share the same exact architecture, which enables elementwise-merging of all adapters in their parameter space. Specifically, given domain-specific adapter weights for all $S$ domains: $\mathcal{A} = \{A_1, \dots, A_S\}$, and input-specific router probabilities $p \in \mathbb{R}^S$, the merged adapter weights $A_M$ are obtained as:

$$A_M = \sum_{s=1}^{S} p_s * A_s. \tag{3}$$

In practice, we only keep the top-$k$ adapters corresponding to the highest router probabilities and set the other probabilities to 0. Our dynamic adapter merging scheme has several benefits. First, it alleviates the impact of incorrect router predictions to improve performance in scenarios where the router fails to produce accurate domain predictions. Second, dynamic adapter merging is a simple, efficient, and effective technique that does not require additional learning processes or costly computational overhead. Third, dynamic adapter merging leverages shared cues from different domains for improved performance in the other domains. Lastly, we note that the commonly used static merging methods (e.g., averaging the weights of all domain-specific models) fail to perform well since the same merged model is used for every single test-time input instance. In comparison, dynamic adapter merging leverages a unique model for every test-time input instance with negligible computational overhead, which improves the model's expressivity and leads to better performance.

Training Sequence: iVQA → MSVD → MSR-VTT → LSMDC → ActivityNet → TGIF

| Method | Downstream VidQA Accuracy (Forgetting) (%) | | | | | | |
|---|---|---|---|---|---|---|---|
| | iVQA | MSVD | MSR-VTT | LSMDC | ActivityNet | TGIF | Avg. |
| Zero-Shot | 26.8 | 33.0 | 15.0 | 51.5 | 25.5 | 41.9 | 32.3 |
| Seq-FT | 28.4 | 36.0 | 23.7 | 52.1 | 31.2 | 67.6 | 39.8 |
| *Multi-task Finetuned (Upper-Bounds)* | | | | | | | |
| Adapters | 39.7 | 56.6 | 46.7 | 62.9 | 42.2 | 67.8 | 52.6 |
| Prompt Tuning | 35.0 | 49.0 | 37.1 | 57.4 | 33.9 | 59.2 | 45.3 |
| *Regularization-based methods* | | | | | | | |
| EwC | 29.9 (-9.9) | 39.3 (-15.5) | 25.5 (-21.2) | 54.9 (-8.1) | 32.4 (-10.0) | 67.5 (-0.5) | 41.6 (-10.9) |
| LwF | 28.3 (-11.5) | 38.2 (-16.6) | 25.8 (-20.9) | 56.4 (-7.6) | 33.6 (-8.8) | **68.5** (+0.5) | 41.8 (-10.7) |
| *Model-merging methods* | | | | | | | |
| Average Merging | 38.0 (-1.8) | 45.7 (-9.1) | 27.7 (-19.0) | 54.5 (-8.5) | 27.0 (-15.4) | 56.6 (-11.4) | 41.6 (-10.9) |
| *Prompt-based methods* | | | | | | | |
| L2P | 32.8 (-2.2) | 43.3 (-5.7) | 32.1 (-5.0) | 54.8 (-3.6) | 27.2 (-6.7) | 54.4 (-4.8) | 40.8 (-4.3) |
| CODA-Prompt | 32.9 (-2.1) | 44.8 (-4.2) | 28.7 (-8.4) | 50.7 (-6.7) | 23.9 (-10.0) | 54.7 (-4.5) | 39.6 (-5.7) |
| S-Prompts | 31.8 (-3.2) | 45.5 (-4.5) | 30.2 (-6.9) | 54.9 (-2.5) | 27.9 (-6.0) | 56.1 (-3.1) | 41.1 (-4.2) |
| DAM | **39.1** (-0.7) | **53.6** (-1.2) | **42.2** (-4.5) | **63.0** (0.0) | **36.3** (-6.1) | 66.8 (-1.2) | **50.2** (-2.3) |

Table 1: Comparison with state-of-the-art on Domain-Incremental VidQA Learning. We individually finetune the adapters and prompts on each dataset, establishing the upper bounds for continual learning methods. We reimplement prior methods using our backbone, as they were not initially designed for VidQA. All continual learning methods are trained sequentially from left to right in the table. Final accuracy is evaluated using the checkpoint trained on the last dataset (TGIF). Our proposed DAM outperforms the current state-of-the-art by 9.1% while exhibiting 1.9% less forgetting.

## 4 EXPERIMENTS

**Datasets and Metrics.** We perform experiments on 6 Video Question Answering (VidQA) datasets, including iVQA (Yang et al., 2021), MSVD-QA (Xu et al., 2017), MSRVTT-QA (Xu et al., 2017), LSMDC (Maharaj et al., 2017), ActivityNet-QA (Yu et al., 2019) and TGIF-QA (Jang et al., 2017). LSMDC is video-conditioned fill-in-blank QA, while the other datasets are open-ended QA. We view **each dataset as a domain** and perform our method on the Continual VidQA setting. Specifically, we sequentially train our method on these domains and evaluate the final checkpoint on all domains with domain identity unknown. Following (Wang et al., 2022c;b), we use the average accuracy and forgetting as the evaluation metrics. Compared to continual learning in image classification tasks, our continual VidQA task poses greater challenges in three aspects: (i) It utilizes six independent VidQA datasets collected at different times by various researchers, resulting in a larger domain gap. (ii) Rather than dividing domain samples equally, the data scale of different domains in our continual VidQA task varies significantly; the largest dataset (LSMDC) contains 48 times the training samples of the smallest one (iVQA). This variability makes the learning process more demanding, yet more representative of real-world scenarios. (iii) VidQA is inherently a more intricate task, necessitating not only visual and textual understanding but also cross-modal reasoning.

**Baselines.** For all of our continual learning baselines (including our approach), we use the state-of-the-art VidQA model FrozenBiLM (Yang et al., 2022), implemented using CLIP ViT-L/14 (Radford et al., 2021) and DeBERTa-V2-XL (He et al., 2020) video and language backbones and containing 1.2B parameters in total. As we are the first to explore DIL for VidQA to our knowledge, we reimplement all the existing methods in our settings (see more details in Appendix A). For our method DAM, we set the temperature $\tau$ to 0.01 and use $k = 2$ for adapter merging. To obtain the upper bound baselines, we separately finetune and evaluate a domain-specific model on each individual dataset (i.e., resulting in 6 domain-specific models for 6 VidQA datasets). For all prior prompt-based approaches, we freeze the pretrained model and add $L = 10$ prompt tokens prepended to their existing tokens as was done in (Wang et al., 2022b). In our comparisons, we include three recent Prompt-based methods L2P (Wang et al., 2022e), CODA-Prompt (Smith et al., 2023a), and S-Prompts (Wang et al., 2022b). We also include two regularization-based methods EwC (Kirkpatrick et al., 2017) and LwF (Li & Hoiem, 2017), both of which, use the same set of adapters and pretrained model as our approach. We report results averaged from 5 runs with different random seeds.

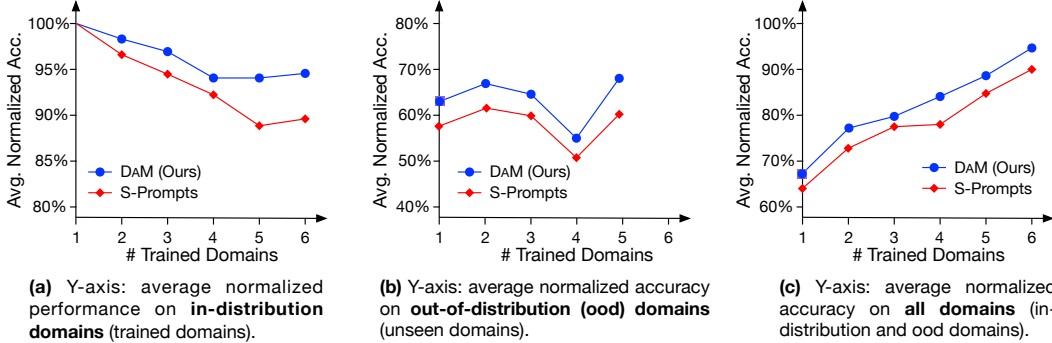

**(a)** Y-axis: average normalized performance on **in-distribution domains** (trained domains).

**(b)** Y-axis: average normalized accuracy on **out-of-distribution (ood) domains** (unseen domains).

**(c)** Y-axis: average normalized accuracy on **all domains** (in-distribution and ood domains).

Figure 3: We study our method's performance on **(a)** in-domain data, **(b)** out-of-domain data, and **(c)** all in-domain and out-of-domain data. We do so by varying the number of trained domains. Normalized Accuracy on each domain is calculated as the model's accuracy divided by the upper bound on that particular domain. We conduct our experiments with 6 domains: iVQA, MSVD, MSR-VTT, LSMDC, ActivityNet, and TGIF. When the number of trained domains is $P$, the remaining $6 - P$ domains are served as out-of-distribution (OOD) domains. For comparison, we use the best-performing prompt-based method, S-Prompts. Our proposed DAM outperforms the S-Prompts for all numbers of trained domains on in- and out-of-domain data.

## 4.1 COMPARISON WITH STATE-OF-THE-ART

Tab. 1 compares our method and state-of-the-art Domain-Incremental Learning (DIL) approaches. Our findings demonstrate that our proposed DAM scheme outperforms the leading DIL method, S-Prompts, by a substantial margin of **9.1%** in average accuracy while also exhibiting **1.9%** less forgetting. Among prompt-based methods, L2P, CODA-Prompt, and S-Prompts show reduced forgetting compared to regularization-based methods EwC and LwF. However, these prompt-based methods achieve even lower accuracy, primarily owing to the constraints of prompt-tuning. Notably, CODA-Prompt's accuracy is on par or lower than L2P and S-Prompts, indicating that joint optimization of the router and prompts is suboptimal for our VidQA task. These results show the effectiveness of the proposed dynamic merging of the continually learned domain-specific adapters.

## 4.2 SCALING THE NUMBER OF DOMAINS

In this subsection, we examine the model's performance as the number of trained domains progressively increases, focusing on both in-distribution and out-of-distribution (OOD) domains. Our evaluation involves comparing our proposed DAM and the top-performing method, S-Prompts. We conduct experiments on datasets spanning six domains: iVQA, MSVD, MSR-VTT, LSMDC, ActivityNet, and TGIF. To ensure comparability across domains, we normalize accuracy for each domain against its respective upper bound baseline (see above).

In Fig. 3, we delve into our method's performance in relation to the number of trained domains. Our observations reveal that DAM consistently surpasses S-Prompts when evaluated on both in-distribution and out-of-distribution (OOD) domains across varying numbers of trained domains. For in-distribution domains, DAM demonstrates a superiority ranging from **1.7%** to **4.8%** in normalized accuracy as the number of trained domains increases from 2 to 6. This emphasizes the scalability of our proposed DAM to effectively accommodate a larger number of domains. Across OOD domains, DAM maintains its advantage by consistently outperforming S-Prompts by **2.9%** to **7.1%** (with an average of **4.8%**) for various numbers of trained domains, signifying enhanced OOD generalization capability. When considering all domains, including in-distribution and OOD, DAM outperforms S-Prompts by an average of **4.0%**. Our DAM exhibits significant advantages over S-Prompts, showcasing its robustness and adaptability in handling domain-incremental VidQA learning scenarios.

## 4.3 ANALYSIS OF THE ROUTER

To study the importance of the router, we experiment with several different router variants. Specifically, we incorporate the router designs from prior methods: L2P (Wang et al., 2022e), S-Prompts (Wang et al., 2022b) and CODA-Prompts (Smith et al., 2023a) into our DAM method.

| Method | Router | Learning | Router Acc (%) | VidQA Acc (%) |
|---|---|---|---|---|
| | Random | - | 16.6 | 40.2 |
| | L2P's | joint | 67.4 | 48.6 |
| DAM | CODA-Prompts' | joint | - | 45.3 |
| | S-Prompts' | disjoint | 76.4 | 49.7 |
| | Ours | disjoint | 79.1 | 50.2 |

Table 2: We study the effectiveness of different router functions. Specifically, we incorporate router functions from several prior methods into our DAM method and measure our model's performance on the downstream VidQA task with each of these routers. We also measure the accuracy of each router function for correctly classifying the domain of a given VidQA input instance. We cannot calculate CODA-Prompts' router's accuracy as it does not explicitly predict the domain identity. From these results, we observe that our non-parametric router function leads to the best downstream VidQA performance despite the simplicity of its design.

| Top-$K$ | MSVD | MSR-VTT | ActivityNet | iVQA | TGIF | LSMDC |
|---|---|---|---|---|---|---|
| 1 (no-merging) | 49.0 | 40.4 | 37.4 | 37.5 | 66.3 | 62.9 |
| 2 | 53.6 | 42.2 | **36.3 (-1.1)** | 39.1 | 66.8 | 63.0 |
| 3 | 54.6 | **42.4 (+2.0)** | 34.0 | 39.3 | **67.0 (+0.7)** | 63.0 |
| 6 (merge all) | **54.9 (+5.9)** | 41.9 | 33.0 | **39.6 (+2.1)** | 66.9 | **63.1 (+0.2)** |
| Router Acc (%) | 51.0 | 69.6 | 76.4 | 81.6 | 96.1 | 100 |

Table 3: We investigate the number of domain-specific adapters to merge for best performance. The Top-$K$ adapters are selected according to the highest router predicted probabilities. The first 4 rows depict the downstream VidQA accuracy, whereas the last row is the router accuracy. We highlight the largest accuracy gap between adapter merging and non-merging variants. Merging adapters is typically useful when the router makes many incorrect predictions.

Our results in Tab. 2 reveal several interesting trends. First, we observe that higher router accuracy typically leads to higher downstream VidQA accuracy, thus indicating the importance of an accurate router function. Second, we notice that jointly training router and domain-specific modules as was done in previous methods (L2P, CODA-Prompt) leads to worse downstream VidQA accuracy than disjoint training (S-Prompts, Ours). Lastly, our results suggest that despite the simplicity of our non-parametric router function, it produces the best performance.

## 4.4 ANALYSIS OF DYNAMIC ADAPTER MERGING

In this section, we analyze the effectiveness of dynamic adapter merging. Specifically, in Tab. 3, we present a comprehensive breakdown of downstream VidQA accuracy and the router's accuracy on each dataset, considering various adapter merging variants. The table highlights an intriguing trend: as the router's accuracy decreases, the benefits derived from model merging become more pronounced. Specifically, when the router's accuracy is at **51.0%** and **69.6%**, adapter merging yields substantial downstream accuracy improvements of **4.9%** and **1.5%** on the MSVD and MSR-VTT datasets, respectively. In contrast, when the router approaches near-perfect accuracy (as seen with a marginal **0.2%** improvement on LSMDC), the gains from adapter merging become less significant. To further validate this observation, Fig. 4 provides insights into the average performance gain of dynamic adapter merging over non-merging variants as a function of router accuracy. The data points are generated by creating a series of routers manually, each predicting domain probabilities with a specified

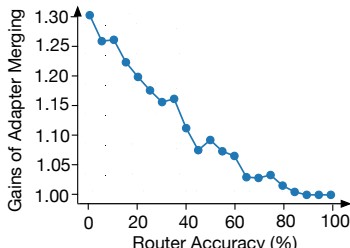

Figure 4: We study the normalized performance gain of dynamic adapter merging as a function of router accuracy. Our results show that dynamic adapter merging leads to a larger boost when the router is inaccurate.

| Method | OK-VQA (test) | aOK-VQA (val) | GQA (val) | VQAv2 (val) | Avg. |
|---|---|---|---|---|---|
| Zero-Shot | 40.7 | 35.7 | 44.0 | 63.1 | 45.9 |
| Ind-FT w/ prompt | 48.2 | 49.3 | 54.4 | 71.3 | 55.6 |
| Ind-FT w/ adapter | 49.2 | 51.8 | 58.7 | 76.2 | 58.8 |
| S-Prompts | 42.9 (-5.3) | 46.1 (-2.2) | 47.3 (-7.1) | 65.3 (-6.0) | 50.4 (-5.2) |
| DAM | **45.1** (-4.1) | **50.4** (-1.4) | **54.1** (-4.6) | **69.8** (-6.4) | **54.8** (-4.0) |

Table 4: We extend our proposed DAM method to continual visual question-answering (VQA) task, utilizing the recent BLIP-2 model (Li et al., 2023b) as the visual-language backbone. We compare the existing state-of-the-art method (S-Prompt) and our DAM with the individual fine-tuning (Ind-FT) results using different parameter-efficient strategies (prompt and adapter). Our method outperforms S-Prompts by 4.8% top-1 accuracy while exhibiting 1.2% less forgetting.

accuracy. The figure confirms the trend observed in Table 3, showcasing that adapter merging offers a **30%** relative improvement when the router's accuracy drops to **0%**.

Based on these results, we can conclude that our proposed adapter merging scheme is particularly advantageous when dealing with many domains. In such complex scenarios, domain prediction becomes notably challenging for the router. This observation aligns seamlessly with our earlier findings (refer to Sec. 4.2), where our method consistently outperforms the previous state-of-the-art to a greater extent when trained on a large number of domains. These collective findings underscore the practical significance and scalability of our proposed approach in real-world domain-incremental VidQA learning scenarios.

## 4.5 EXTENSION TO VISUAL QUESTION-ANSWERING

To show the flexibility of the proposed DAM approach, we extend it to a visual (image) question-answering (VQA) task. We integrate our proposed DAM and the best performing prompt-based baseline S-Prompts with the state-of-the-art VQA model, BLIP-2 (Li et al., 2023a), which uses CLIP ViT-G/14 (Radford et al., 2021) and FlanT5-XL (Chung et al., 2022) as its vision-language backbone and has 4.1B parameters in total. We then continually train both models on 4 mainstream VQA datasets: OK-VQA (Marino et al., 2019), aOK-VQA (Schwenk et al., 2022), GQA (Hudson & Manning, 2019) and VQAv2 (Goyal et al., 2017). The results are shown in Tab. 4. Our proposed DAM outperforms S-Prompts by **4.4%** with **1.2%** less forgetting, thus, demonstrating the generality of our approach beyond the video-level settings.

## 5 DISCUSSION AND CONCLUSION

In this work, we investigate rehearsal-free domain-incremental VidQA learning by combining continually learned domain-specific adapters and model merging techniques. We outperform existing state-of-the-art by **9.1%** with **1.9%** less forgetting on a benchmark with six distinct video domains. The proposed method DAM is simple and flexible, and we further extend it to visual question-answering using a 4B parameter model BLIP-2, demonstrating our method's generalization beyond video-level scenarios. Despite effective results, we also observe a few limitations of our proposed approach. Firstly, our approach employs a straightforward weighted averaging technique for merging adapter weights, leaving room for more advanced merging methods that could enhance knowledge sharing among domains and further improve performance. Secondly, our validation encompasses a relatively small number of domains (six in our case), consistent with previous domain-incremental learning research. It would be valuable to assess the effectiveness of our method and existing domain-incremental learning methods across a more extensive domain spectrum, potentially involving a substantial number of domains (e.g., 100). In future research, we aspire to extend our approach to other tasks, including image classification and video classification. We believe that our exploration and analysis of router and model merging techniques can serve as valuable insights for both the model merging and continual learning communities, inspiring further advancements in these domains.

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

## APPENDIX

In this appendix, we present the following:

A. Implementation Details.

B. Efficiency Analysis of Adapters.

C. DAM as a model merging technique.

D. Generalizability studies.

E. Additional ablations.

F. Comparison with In-Context Learning.

G. Dataset Descriptions.

## A  IMPLEMENTATION DETAILS

**Details of our DAM approach.** Our choice for the VidQA model is FrozenBiLM (Yang et al., 2022), a state-of-the-art (SOTA) model in the VidQA domain. To align with this model, we utilize a vocabulary encompassing the 3635 most frequent answers. Adhering to the FrozenBiLM approach, we integrate adapters into each layer of the DeBERTa-XL (He et al., 2020) language model, employing a downsampling rate of 8. The loss function is the same as the original FrozenBiLM model, i.e., the cross-entropy loss between the predicted tokens and ground-truth answer tokens. For the initialization of domain-specific adapters during the commencement of continual learning (first domain), we use the weights from FrozenBiLM, which is pre-trained on a substantial dataset comprising 10 million video-text pairs (WebVid10M (Bain et al., 2021)). In the training of domain-specific adapters for each subsequent domain, we conduct 20 epochs of training with an initial learning rate of $5e-5$. The learning rate undergoes a linear warm-up for the first 2 epochs, followed by a linear decay to 0. Our proposed DAM introduces only two hyper-parameters. Specifically, we set the temperature parameter ($\tau$) to 0.01 and employ a value of $k = 2$ for the adapter merging process.

**Network Structures.** The base model is FrozenBiLM Bain et al. (2021). The pretrained model $f$ (Sec. 3.2) is the pretrained FrozenBiLM except that we concat the averaged hidden states from the 4th last layer of the DeBERTa-XL and the averaged hidden states from the last layer of CLIP-L/14 as the output. Following (Yang et al., 2022; Houlsby et al., 2019; Yang et al., 2022), the adapters in our approach consist of a downsampling and an upsampling linear layer, along with a residual connection. The linear layers are set with an 8x downsample scale to intermediate hidden size and the upsampler maps back to the original dimensionality.

**Continual Learning Baselines.** Since our work is the very first exploration of continual VidQA learning, we implement a number of continual learning baselines (focused on image classification) to VidQA task, including three recent Prompt-based methods L2P (Wang et al., 2022e), CODA-Prompt (Smith et al., 2023a), and S-Prompts (Wang et al., 2022b)and two regularization-based methods EwC (Kirkpatrick et al., 2017) and LwF (Li & Hoiem, 2017). For a fair comparison, we use the same pretrained model and preserve most hyper-parameter settings with our approach.

- **L2P** (Wang et al., 2022e). For the prompt settings, we set the prompt length to 10 and the size of the prompt pool to 6. The dimension of the prompt key is configured to be 3072, matching the dimension of the router input feature in our method. The prompt dimension is set to 1536, aligning with the input dimension of the frozen language model. We sweep the learning rate between $1e-2$ and $1e-5$ with an interval of $3.33\times$. The best performance is achieved with an initial learning rate 3e-3.

- **CODA-Prompt** (Smith et al., 2023a). For a fair comparison, we adopt the same prompt settings as our L2P baseline for CODA-Prompt. Following Smith et al. (2023a), we apply orthogonality initialization to initialize the prompts, their keys, and their attention matrices. The dimension of prompt attention is set to 3072, consistent with the dimension of the prompt key. For optimal performance, we configure the learning rate to 1e-3.

- **S-Prompts** (Wang et al., 2022b). We use exactly the same prompt settings as in our implementation for L2P. For their K-Means router, we set $K = 3$ as the number of centroids for each domain and 1-first-nearest neighbor with the centroids to search for the best prompts.

- **EwC** (Kirkpatrick et al., 2017) and **LwF** (Li & Hoiem, 2017). We follow their original implementations, except that the regularization is only applied to adapters as all the other parameters are frozen. The same set of adapters are used for all the domains.

## B  EFFICIENCY ANALYSIS OF ADAPTERS

The adapters introduced in each domain contribute merely **2.5%** of the pretrained model's parameters (CLIP-L/14 + DeBERTa-V2-XLarge), totaling 30M parameters. With 10 domains, this results in only a **25%** increase in parameters, a reasonable augmentation given DAM's robust performance. In terms of computational cost, merging adapter parameters incurs just **0.09** GFLOPs (30M *(2k-1), k=2 in our case), notably lower than the **162** GFLOPs required by CLIP-L/14 for a single image processing. We appreciate the reviewer's suggestion and will integrate this analysis into the revision.

| Method | iVQA | MSVD | MSR-VTT | LSMDC | ActivityNet | TGIF | Avg. |
|---|---|---|---|---|---|---|---|
| Multi-task (upper-bound) | 39.7 | 56.6 | 46.7 | 62.9 | 42.2 | 67.8 | 52.6 |
| Avg. Merging | **38.0** | 45.7 | 27.7 | 54.5 | 27.0 | 56.6 | 41.6 |
| RegMean | 36.6 | 49.7 | 32.5 | 54.0 | 27.7 | 57.8 | 43.1 |
| DAM (Ours) | 36.5 | **51.6** | **39.5** | **63.0** | **36.5** | **67.7** | **49.1** |

Table 5: Comparision with existing model merging techniques. All the methods merge the same set of domain models that are individually finetuned on each dataset.

| Method | MSVD | iVQA | LSMDC | ActivityNet | AGQA | Env-QA | TrafficQA($\frac{1}{2}$) | Avg. |
|---|---|---|---|---|---|---|---|---|
| Upper-Bound | 56.6 | 39.7 | 63.0 | 42.2 | 63.4 | 32.3 | 67.8 | 52.1 |
| DAM | 54.0 | 39.3 | 63.0 | 37.4 | 63.3 | 32.0 | 67.8 | 51.0 |
| Router Acc. of DAM | 60.7 | 83.7 | 100 | 78.4 | 99.9 | 99.2 | 99.7 | 88.7 |

Table 6: Domain-Incremental Learning (DIL) on 7 diverse domains.

## C  DAM AS A MODEL MERGING TECHNIQUE

The proposed method DAM can also be used as a model merging technique. The concept of dynamic merging could be inspiring to the model merging community. As shown in Tab. 5, we compare DAM with the other merging approaches, including average merging and RegMean Jin et al. (2022), while all the approaches merge the same set of domain models that are individually finetuned on each dataset. Unlike the other approaches, DAM determines the merging ratios for domain adapters based on the input instance, and this flexibility makes DAM outperform RegMean by **6.0%** and average merging by **7.5%** in average accuracy. The results show the potential of the proposed selective and dynamic merging strategy to inspire model-merging communities.

## D  GENERALIZABILITY STUDIES

We showcase the generalizability of the proposed DAM by further experimenting with **Domain-Incremental Learning (DIL)** on more diverse domains, as well as **Class-Incremental Learning (CIL)** and **Task-Incremental Learning (TIL) scenarios**.

**Domain-Incremental Learning on more diverse domains (DIL).** We experiment on more diverse VidQA domains. This setup incorporates 4 originally benchmarked domains: social videos (MSVD), instructional videos (iVQA), movie videos (LSMDC), and long activity videos (ActivityNet). Additionally, we introduce 3 new domains: indoor human videos (AGQA), traffic videos (TrafficQA), and virtual videos (Env-QA). Tab. 6 shows DAM's answer prediction accuracy and the router's domain identity prediction accuracy. DAM only has -1.1% forgetting on this setting, which is even 1.2% less than on our original 6-domain setting. This is because the router performs better on the domains (e.g. AGQA, EnvQA, TrafficQA) that are significantly different from the others. The proposed DAM is better at dealing with more diverse domains as they are easier to distinguish by the router function.

**Class-Incremental Learning (CIL).** We treat each unique answer as a class and adhere to a protocol commonly employed in continual image classification (Wang et al., 2022e). Specifically, we experiment on two settings:1) **10 tasks split from MSRVTT-QA** with non-overlapping classes between tasks, and 2) **4-Datasets** (iVQA, MSVD, LSMDC, ActivityNet), excluding samples with overlapping answers across datasets. The results, presented in Tab. 7, indicate that DAM consistently outperforms S-Prompts (Wang et al., 2022b), achieving 18.2% and 8.5% improvement on average accuracy on MSRVTT-QA 10-tasks and 4-Datasets respectively. Note that S-Prompts is compared with prompt-based multi-task finetuning when calculating the forgetting.

The CIL evaluation mimics the evaluation of the out-of-date issue as shown in Fig. 1. The old model may not be able to answer questions in new tasks as they never see the classes in the new tasks before, which is similar to the example in Fig. 1 that a VidModel trained in 2021 may struggle with questions about the 2023 movie "Barbie".

| Method | MSRVTT-QA 10-tasks | | 4-Datasets | |
|---|---|---|---|---|
| | Average Acc. | Forgetting | Average Acc. | Forgetting |
| Upper-Bound | 47.3 | - | 51.6 | - |
| S-Prompts | 15.4 | -23.5 | 42.2 | -3.3 |
| DAM (Ours) | 33.6 | -13.7 | 50.7 | -0.9 |

Table 7: Class-Incremental Learning (CIL) on two settings:1) 10 tasks split from MSRVTT-QA, and 2) 4-Datasets (iVQA, MSVD, LSMDC, ActivityNet).

| Method | iVQA | MSVD | MSR-VTT | LSMDC | ActivityNet | TGIF | Avg. |
|---|---|---|---|---|---|---|---|
| Upper-Bound | 39.7 | **56.6** | 46.7 | 62.9 | 42.2 | 67.8 | **52.6** |
| DAM | **39.8** | 54.8 | **46.7** | **63.0** | **42.4** | **68.0** | 52.5 |

Table 8: Application to Task-Incremental Learning (TIL).

**Task-Incremental Learning (TIL).** We treat each dataset as a task. Unlike DIL or CIL, TIL is provided with task indexes for inference and thus DAM using task-specific adapters could overcome forgetting, i.e. DAM will always use the adapters of the task that the testing instance belongs to.

# E  ADDITIONAL ABLATIONS

**Order of domains.** In Tab. 9, we study how the order of domains affects our model's performance. We randomly sample 5 domain orders and train our framework using those orders. Based on the results in the table, we observe that the performance of our approach is quite stable across all 5 domain orders (**50.56 ± 0.26%**). This indicates our method is insensitive to the order of domains.

**Effectiveness of continual initialization.** In Section 3.1, we introduced a continual initialization scheme for initializing a current domain-specific adapter using the weights of a previously learned adapter. In Tab. 10, we validate the effectiveness of this scheme and show that it leads to a notable **1.1%** average accuracy improvement. These improvements are particularly pronounced for the domains that are trained first, such as iVQA and MSVD. We posit that the benefits of continual initialization stem from the fact that the weights of continually learned adapters reside in a more similar parameter space. This phenomenon contributes to reducing interference disagreements when merging adapters, as discussed in Yadav et al. (2023).

**Design choice of router function** Besides the router function used by existing work, including MLP-based learnable routers proposed by L2P and CODA-Prompts, as well as the KMeans-based router employed in S-Prompts. We also tried more advanced router functions on the proposed DAM framework. As shown in Tab. 11, their performance is only comparable to our router. Thus, we keep our router simple but effective in the DAM approach..

| Domain Order | Avg. Acc (%) |
|---|---|
| V D T L A G | 50.2 |
| L T G D A V | 50.8 |
| V A D G T L | 50.4 |
| G T A V D L | 50.9 |
| V A G D T L | 50.5 |

Table 9: Ablations on the order of domains. We randomly sampled 5 orders and obtained stable average accuracies (50.56±0.26%). V: iVQA; D: MSVD; T: MSR-VTT; L: LSMDC; A: ActivityNet; G: TGIF.

| Method | iVQA | MSVD | MSR-VTT | LSMDC | ActivityNet | TGIF | Avg. |
|---|---|---|---|---|---|---|---|
| DAM | **39.1** | **53.6** | **42.2** | **63.0** | 36.3 | 66.8 | **50.2** |
| w/o continual initialization | 36.5 | 51.6 | 39.5 | **63.0** | **36.5** | **67.7** | 49.1 |

Table 10: DAM benefits from the proposed continual initialization.

| Router | iVQA | MSVD | MSR-VTT | LSMDC | ActivityNet | TGIF | Avg. |
|---|---|---|---|---|---|---|---|
| GMM | 38.5 | 55.1 | 43.4 | 63.0 | 31.2 | 65.4 | 49.4 |
| Learnable MLP | 39.1 | 49.9 | 42.9 | 63.0 | 31.1 | 67.4 | 48.9 |
| Ours (cos. sim.) | 39.1 | 53.6 | 42.2 | 63.0 | 36.3 | 66.8 | 50.2 |

Table 11: Design choices of router function.

## F COMPARISON WITH IN-CONTEXT LEARNING

Few-shot in-context learning (ICL) can be another approach to address catastrophic problem in continual learning. We further experiment with the one-shot in-context learning using FrozenBiLM and report the results below. The proposed DAM outperforms one-shot FrozenBiLM by **29.3%** in average accuracy. The inferior performance of one-shot ICL is because LLM with at least **6.7B** parameters begin to have in-context learning ability on multimodal tasks (Koh et al., 2023).

## G DATASET DESCRIPTIONS

**Video Question Answering(VidQA).** We evaluate our model on 5 open-ended VidQA datasets iVQA (Yang et al., 2021), MSVD-QA (Xu et al., 2017), MSRVTT-QA (Xu et al., 2017), ActivityNet-QA (Yu et al., 2019) and TGIF-QA (Jang et al., 2017) and a video-conditioned fill-in-the-blank dataset LSMDC-FiB (Maharaj et al., 2017).

- **iVQA** (Yang et al., 2021) is an open-ended VidQA dataset with reduced language biases and high-quality redundant manual annotations. It contains 10K video clips and 10K questions, split into 6K/2K/2K for training/validation/testing.
- **MSVD-QA** (Xu et al., 2017) is an open-ended VidQA dataset based on Microsoft Research Video Description Corpus (Chen & Dolan, 2011). It contains 1.8K video clips and 51K question-answer pairs, split into 32K/6K/13K for training/validation/testing.
- **MSRVTT-QA** (Xu et al., 2017) is an open-ended VidQA dataset based on MSR-VTT dataset (Xu et al., 2016). It contains 10K video clips and 243K question-answer pairs, split into 158K/12K/73K for training/validation/testing.
- **ActivityNet-QA** (Yu et al., 2019) is an open-ended VidQA dataset based on long videos (Caba Heilbron et al., 2015) (averaging 180 seconds) and human annotation. It contains 5.8K video clips and 58K question-answer pairs, split into 32K/18K/8K for training/validation/testing.
- **TGIF-QA** (Jang et al., 2017) is an open-ended VidQA dataset based on the Tumblr GIF (TGIF) dataset (Li et al., 2016). It contains 46K GIFs and 53K question-answer pairs, split into 39K/13K for training/testing.
- **LSMDC-FiB** (Maharaj et al., 2017) is an open-ended video-conditioned fill-in-the-blank task that consists of predicting masked words in sentences that describe short movie

| Method | iVQA | MSVD | MSR-VTT | LSMDC | ActivityNet | TGIF | Avg. |
|---|---|---|---|---|---|---|---|
| Zero-Shot | 26.8 | 33.0 | 15.0 | 51.5 | 25.5 | 41.9 | 32.3 |
| One-Shot ICL | 17.9 | 22.5 | 9.7 | 34.5 | 17.8 | 23.1 | 20.9 |
| Ours (cos. sim.) | **39.1** | **53.6** | **42.2** | **63.0** | **36.3** | **66.8** | **50.2** |

Table 12: Comparison with few-shot in-context learning.

clips (Rohrbach et al., 2015). It contains 119K video clips and 349K question-answer pairs, split into 297K/22K/30K for training/validation/testing.

**Visual Question Answering(VQA).** Follow Li et al. (2023b), we evaluate our model on 4 mainstream VQA datasets: OK-VQA (Marino et al., 2019), aOK-VQA (Schwenk et al., 2022), GQA (Hudson & Manning, 2019) and VQAv2 (Goyal et al., 2017).

- **OK-VQA** (Marino et al., 2019) is a knowledge-based visual question-answering benchmark with 14k images and 14k questions.
- **aOK-VQA** (Schwenk et al., 2022) is an augmented successor of OK-VQA (Marino et al., 2019) and contains a diverse set of 25K questions requiring a broad base of commonsense and world knowledge to answer.
- **GQA** (Hudson & Manning, 2019) is a large-scale visual question-answering dataset with real images from the Visual Genome (Krishna et al., 2017) dataset and balanced question-answer pairs.
- **VQAv2** (Goyal et al., 2017) consists of 1.1M questions about COCO images (Chen et al., 2015) each with 10 answers. It is the balanced version of the original VQA (Antol et al., 2015) dataset.

