# OpenReview forum: "Dynamic Adapter Merging for Continual Video Question-Answering Learning"
_ICLR.cc/2024/Conference — Submitted to ICLR 2024_

### Official Review · Reviewer_yoGd · 2023-10-19

**Soundness:** 3 good
**Presentation:** 2 fair
**Contribution:** 2 fair
**Rating:** 5
**Confidence:** 5

**Summary:**

This paper proposes a dynamic adapter merging framework for domain-incremental VideoQA learning. The framework is capable of obtaining multiple domain-specific adapters and dynamically integrating different domain information through model merging techniques. Experiments results on multiple public datasets verify the effectiveness of the proposed method.

**Strengths:**

1.	The logic of the paper is reasonable.
2.	The experiments are relatively adequate.

**Weaknesses:**

The technical details of this paper are not described clearly enough, my concerns are as follows:
1.	Why do you set up N adapters for each domain instead of one?
2.	Why do you choose to insert domain-specific adapters after the self-attention and feed-forward layers, respectively? What are the considerations?
3.	What exactly is meant by the pre-trained model f in Eqn. (1)?
4.	What does the symbol k in the baselines section on page 6 refer to? I cannot find a definition in the previous text.
5.	What is the exact structure of the adapter?

**Questions:**

See above.

---

> ### Author Response · Authors · 2023-11-19
> **Official Comment by Authors**
>
> We acknowledge the oversight in omitting specific details, including the adapter structure and design rationales, assuming a shared background with the readers. In response to your valuable feedback, we recognize the need to enhance accessibility for a broader audience. More implementation details will be meticulously incorporated into both the main paper and supplementary materials.
>
> ---
>
> > **C4.1 Why do you set up N adapters for each domain? Why insert these adapters after the self-attention and feed-forward layers, respectively?**
>
> We regret any confusion. Given that adapter design is not the primary focus of our work, we adhere to recent literature  [1,2,3] in incorporating multiple adapters (N) into the pretrained model for multimodal applications. These adapters are typically inserted after the self-attention and feed-forward layers in Transformer-based architectures. The primary objective of employing multiple adapters is to augment the model's capacity with minimal impact on computational cost and parameters. This design choice allows for the dynamic merging of expert models for each test instance with negligible overhead, as detailed in Section C3.3 of the rebuttal.
>
> ---
>
> > **C4.2 What is the exact structure of the adapter?**
>
> Following previous works [1,2,3], the adapters in our approach consist of a downsampling and an upsampling linear layer, along with a residual connection. The linear layers are set with an 8x downsample scale to intermediate hidden size and the upsampler maps back to the original dimensionality [1].
>
> [1]: Sung, Yi-Lin et al. "Vl-adapter: Parameter-efficient transfer learning for vision-and-language tasks." CVPR 2022.
> [2]: Houlsby, Neil, et al. "Parameter-efficient transfer learning for NLP." ICML 2019.
> [3]: Yang, Antoine, et al. "Zero-shot video question answering via frozen bidirectional language models." NeurIPS 2022.
>
> ---
>
> > **C4.3 What exactly is meant by the pre-trained model f in Eqn. (1)?**
>
> We refer to the original pretrained model as “f”, i.e. the model used for zero-shot inference (FrozenBiLM in our case) except that we extract the hidden states from the 4th last layer as the output. We thank the reviewer’s suggestion and will add a detailed description in the supplementary materials.
>
> ---
>
> > **C4.4 What does the symbol k in the baselines section on page 6 refer to?**
>
> We regret any confusion caused by the omission of the definition of the symbol 'k' in our manuscript. In our context, 'k' denotes that only the top-k models participate in the merging process (as described in Section 3.3). The rationale behind selecting only the top-k models is to filter out irrelevant adapters, while still capitalizing on the advantages of adapter merging. Additionally, maintaining a fixed 'k' ensures a constant computational cost, regardless of the number of domains. We appreciate the reviewer's understanding, and we will include this clarification in the revised manuscript.
>
> ---
>
> Besides the implementation details, please take a look at our **General Response** above to see more about experimental setup and generalizability. We are happy to have more in-depth discussions with you if you have further concerns!
>
> *If our response addresses your concerns, please consider increasing the scores. Also, feel free to ask follow-up questions during the rebuttal period.*

---

### Official Review · Reviewer_DkBm · 2023-11-04

**Soundness:** 2 fair
**Presentation:** 3 good
**Contribution:** 3 good
**Rating:** 6
**Confidence:** 4

**Summary:**

The paper studies VideoQA in a domain continual-learning setting. The task encourages VQA models that can quickly adapt to new domains/datasets while simultaneously prevent catastrophic forgetting on learned domains. To achieve the goal, the paper proposes the dynamic adapter merging (DAM) method. Given a random instance, DAM dynamically (learning-free) merges a series of domain-specific parameter adapters for answer prediction, where the adapters are continually learned across datasets of different domains. The authors conduct extensive experiments on 6 VideoQA datasets and additionally 4 ImageQA datasets to show the effectiveness of the proposed method.

**Strengths:**

1.	The paper conducts the first study on domain-incremental learning in VideoQA. It also presents a nice solution to benchmark the task.
2.	The DAM method is simple, easy to understand and shows strong results as well. Also, the experiments and analyses are in-depth.
3.	The paper is well-presented and easy to read.

**Weaknesses:**

1.	The definition of domain regarding VideoQA is not clear. The authors simply treat different datasets as different domains. This is certainly problematic and prevents detailed model analysis. For example, regarding the question type, all datasets define similar questions except for LSMDC with fill-in-blank setting.  Regarding the video type, there are instructional videos (iVQA), social videos (MSVD, MSRVTT, TGIF), movie videos (LSMDC) and activity videos. Regarding video length, all videos are short (3~15s) except for ActivityNet(3 mins). It would be better to experiment with more clarified domains instead of datasets.

2.	While the ‘dynamic merging’ design mitigates the problem of catastrophic forgetting and improves the overall performance as well, it necessitates all the learned adapters for inference. This resembles more on model ensemble versus continual learning a ‘single’ model. It is necessary to show the size of the adapters and analyze the efficiency.

3.	The authors obtain the upper-bound results by individually finetuning on target datasets. My concern is that this ‘upper-bound’ may not be the actual upper-bound for incremental-learning because of data augmentation. Moreover, the gap between DAM' results and this upper-bound results is too small to show that there is need for future efforts as a novel task setting. The authors need to find a more convincing upper-bound or just mention the current one as a reference.

4.	According to the task setting, providing more analyses /comparisons on an OOD setting (aside from Fig.3(b)) would make the experiments more sound.

**Questions:**

Minor:
1. Why is the performance on ActivityNet not as good as on other datasets?

2. In Sec. 3.2, what specific model does ‘f’ refer to?

3. Analyses of table 5, 6 should be moved from the appendix to the main text.

---

> ### Author Response · Authors · 2023-11-19
> **Official Comment by Authors**
>
> We appreciate your insightful review and constructive suggestions. In response, we have diligently addressed all raised concerns, and we believe our revisions effectively resolve them.
>
> ---
>
> > **C3.1 The definition of domain regarding VideoQA is not clear.**
>
> Our definition of the domain diverges from the conventional definition in the continual learning community. Please refer to the **“Definition of domains” of G1 in General Response** for more detail.
>
> We also report the results on 7 diverse VidQA domains (separated by video type) in **“DIL on more diverse domains” of G2 in General Response**.
>
> ---
>
> > **C3.2 It would be better to experiment with more clarified domains instead of datasets.**
>
> We further validate our method DAM on **7 more diverse domains**: social videos (MSVD), instructional videos (iVQA),  movie videos (LSMDC), long activity videos (ActivityNet), indoor human videos (AGQA), traffic videos (TrafficQA) and virtual videos (Env-QA). Results show that DAM has even less forgetting on more clarified domain settings.
>
> Please refer to **"DIL on more diverse domains" of G2 in General Response** for more details.
>
> ---
>
> > **C3.3 The proposed DAM necessitates all the adapters for inference. It is necessary to analyze the efficiency.**
>
> The adapters introduced in each domain contribute merely **2.5%** of the pretrained model's parameters (CLIP-L/14 + DeBerTa-V2-XLarge), totaling **30M** parameters. With 10 domains, this results in only a 25% increase in parameters, a reasonable augmentation given DAM's robust performance.
>
> In terms of computational cost, merging adapter parameters incurs just **0.09 GFLOPs** (30M *(2k-1), k=2 in our case), notably lower than the **162 GFLOPs** required by CLIP-L/14 for a single image processing. We appreciate the reviewer's suggestion and will integrate this analysis into the revision.
>
> ---
>
> > **C3.4 The authors obtain the upper-bound results by individually finetuning target datasets. My concern is that this ‘upper-bound’ may not be the actual upper-bound for incremental learning because of data augmentation.**
>
> We conducted both individual finetuning (Ind-FT) and multi-task finetuning (MLT-FT), the latter involving joint training on all datasets. As indicated in the table, MLT-FT yields a notable 1.8% enhancement in MSVD-QA, while maintaining a comparable average accuracy to Ind-FT (**52.5% vs. 52.6%**).
>
> In the realm of continual learning, multi-task finetuning is conventionally considered an upper bound. We apologize for the confusion and will replace the upper bound with MLT-FT in the revised manuscript.
>
> |Method|iVQA|MSVD|MSR-VTT|LSMDC|ActivityNet|TGIF|Avg.|
> |-|-|-|-|-|-|-|-|
> |Ind-FT|39.8|54.8|46.7|63.0|42.4|68.0|52.5|
> |MLT-FT|39.7|56.6|46.7|62.9|42.2|67.8|52.6|
>
> ---
>
> > **C3.5 The gap between DAM's results and these upper bounds is too small to show the need for future efforts in a novel task setting.  Also, why is the performance on ActivityNet not as good as others?**
>
> We thank the reviewer for pointing out the room for future work. There are still unaddressed issues that lead to a relatively large accuracy drop on some specific datasets, i.e. MSRVTT (**-4.5%**) and ActivityNet (**-6.1%**). Other future work includes experimenting with **many more domains (or datasets) like 100**, and extending our method to extremely large models (i.e. > **10B** parameters).
>
> Regarding the ActivityNet dataset, we analyze the performance drop might be due to the high similarity of its question type with the iVQA dataset, while the videos are quite different. This inconsistency may lead to confusion within our proposed model.
>
> ---
>
> > **C3.6 Providing more analyses /comparisons on an OOD setting (aside from Fig.3(b)) would make the experiments more sound.**
>
> We further experiment on 7 diverse domains as described in G2 of the General Response. We continually train the proposed DAM on the first 3 domains and evaluate the trained model on the other 3 domains. From the table, DAM outperforms S-Prompts consistently in all the domains, attributed to the larger capacities of adapters and the robustness of dynamic merging.
>
> ||In Distribution|||Out-of-Distribution|||
> |-|-|-|-|-|-|-|
> |Method|MSVD|iVQA|LSMDC|ActivityNet|AGQA|Env-QA|TrafficQA|
> |Upper-Bound|56.6|39.7|63.0|42.2|63.4|32.3|67.8|
> |DAM|54.4|39.4|63.0|20.3|25.4|5.6|18.9|
> |S-Prompts|47.7|34.0|57.4|18.7|22.9|5.5|15.1|
>
> ---
>
> > **C3.7 In Sec. 3.2, what specific model does ‘f’ refer to?**
>
> We refer to the original pretrained model as “f”, i.e. the model used for zero-shot inference (FrozenBiLM in our case) except that we extract the hidden states from the 4th last layer as the output. We thank the reviewer’s suggestion and will add a detailed description in the supplementary materials.
>
> ---
>
> *If our response addresses your concerns, please consider increasing the scores. Also, feel free to ask follow-up questions during the rebuttal period.*

---

### Official Review · Reviewer_thbM · 2023-11-04

**Soundness:** 3 good
**Presentation:** 3 good
**Contribution:** 2 fair
**Rating:** 5
**Confidence:** 4

**Summary:**

The article presents to address continual video question-answering (VidQA) learning with a simple framework, named DAM. Through sequentially training domain-specific adapters and leveraging a video-language router to merge the adapters for inference, DAM outperforms prior methods by 9.1% while forgetting less by 1.9%.

**Strengths:**

1) The paper assumes that this is the first attempt to address the issue of continual learning in VideoQA.
2) Comprehensive Ablation Studies: The article includes sufficient and in-depth set of ablation experiments, which provide a thorough understanding of the method's performance and help identify critical components.
3) Clear Method Framework: The method's framework is straightforward and well-explained, making it accessible to readers and researchers in the field.

**Weaknesses:**

1) Limited Dataset Diversity: The article's experimental use of six datasets with relatively small differences between them, especially MSVD and MSR-VTT, raises concerns about the method's domain adaptation and continual learning capabilities. The use of internet-sourced videos in the datasets does not fully explore the potential challenges posed by more diverse datasets, such as those collected in virtual environments (e.g., Env-QA[1]), traffic scenarios (e.g., TrafficQA[2]), or indoor human activities (e.g., AGQA[3]). What’s more, the out-of-date issue proposed in Figure 1 hasn’t been evaluated, also.
2) While the article demonstrates the effectiveness of the adapter and router, their simple design might not generalize well to more challenging datasets. The reviewer has doubts about their applicability in more complex scenarios.
3) The article does not provide a fair comparison with backbone models under few-shot learning setting. A direct comparison between in-context learning using FrozenBiLM and the proposed approach could offer a more comprehensive evaluation.
[1] Gao, Difei, et al. "Env-qa: A video question answering benchmark for comprehensive understanding of dynamic environments." Proceedings of the IEEE/CVF International Conference on Computer Vision. 2021.
[2] Xu, Li, He Huang, and Jun Liu. "Sutd-trafficqa: A question answering benchmark and an efficient network for video reasoning over traffic events." Proceedings of the IEEE/CVF Conference on Computer Vision and Pattern Recognition. 2021.
[3] Grunde-McLaughlin, Madeleine, Ranjay Krishna, and Maneesh Agrawala. "Agqa: A benchmark for compositional spatio-temporal reasoning." Proceedings of the IEEE/CVF Conference on Computer Vision and Pattern Recognition. 2021.

**Questions:**

1) The article does not provide sufficient evidence of severe catastrophic forgetting in current large models.
2) It is worth discussing whether there are unique challenges related to continual learning in the domain of VideoQA.

---

> ### Author Response · Authors · 2023-11-19
> **Official Comment by Authors**
>
> We appreciate your insightful review and constructive suggestions. In response, we have diligently addressed all raised concerns, and we believe our revisions effectively resolve them.
>
> ---
>
> > **C2.1 Limited Dataset Diversity: The article uses 6 datasets with relatively small differences between them, especially MSVD and MSR-VTT.**
>
> We thank the reviewer for the valuable suggestion. We believe the confusion is caused by the question of **whether domains should be orthogonal**.
>
> We elaborate on our point in “**Dataset Diversity” of  G1 in the General Response** above, please refer to that section for more detail. We also report the results with more diverse datasets in **G2 in the General Response**.
>
> ---
>
> > **C2.2 The out-of-date issue proposed in Fig. 1 hasn’t been evaluated.**
>
> We further evaluate the CIL scenario to mimic the evaluation of the out-of-date issue. The old model may not be able to answer questions in new tasks as they never see the classes in the new tasks before, which is similar to the example in Fig. 1 that a VidModel trained in 2021 may struggle with questions about the 2023 movie “Barbie”. Our proposed DAM method archives significant improvements compared to the second-best approach on CIL settings.
>
> We refer the reviewer to **G2 (Generalization) in the General Response** for more details.
>
> ---
>
> > **C2.3 The use of internet-sourced videos in the datasets does not fully explore the potential challenges posed by more diverse datasets, such as those collected in virtual environments (Env-QA), traffic scenarios (TrafficQA), or indoor human activities (AGQA).**
>
> Thanks for the insightful suggestion. We further experiment with **7 diverse video datasets** including all the datasets that the reviewer mentioned. Results show that our DAM method achieves even less forgetting on more diverse datasets.
>
> Please refer to “**DIL on more diverse domains” of G2 in General Response** with more detail.
>
> ---
>
> > **C2.4 Their simple design might not generalize well to more challenging datasets. The reviewer has doubts about this.**
>
> In **G2 Generalization of the General Response**, we showcase DAM’s generalization ability by experimenting on DIL with more diverse domains, CIL, and TIL scenarios. In Sec 4.5 and Tab. 4 in the main paper, we show DAM can be applied to the visual (image)-QA task using a different large model BLIP2.
>
> All the experimental results support the conclusion that DAM can generalize well to challenging datasets, scenarios, and even more tasks and models. This is because we do not need complex design on top of pretrained large models as they already provide enough generalizability.
>
> ---
>
> > **C2.5 The article does not provide a fair comparison with few-shot learning. A direct comparison between in-context learning using FrozenBiLM and the proposed approach could offer a more comprehensive evaluation.**
>
> We appreciate that the reviewer pointed out another interesting direction (in-context learning, ICL) to address the continual learning problem. We further experiment with the one-shot in-context learning using FrozenBiLM and report the results below. The proposed DAM outperforms one-shot FrozenBiLM by **29.3%** in average accuracy. The inferior performance of one-shot ICL is because LLM with at least 6.7B parameters **begin to have** in-context learning ability on multimodal tasks (Koh et al. 2023).
>
> We will add this comparison and mention that ICL with LLM could be a potential direction for continual learning in the revised manuscript.
>
> |Method|iVQA|MSVD|MSR-VTT|LSMDC|ActivityNet|TGIF|Avg.|
> |-|-|-|-|-|-|-|-|
> |Zero-shot FrozenBiLM|26.8|33.0|15.0|51.5|25.5|41.9|32.3|
> |One-shot ICL FrozenBiLM|17.9|22.5|9.7|34.5|17.8|23.1|20.9|
> |DAM(Ours)|39.1|53.6|42.2|63.0|36.3|66.8|50.2|
>
> Koh, Jing Yu et al. "Grounding Language Models to Images for Multimodal Inputs and Outputs." (2023).
>
>
> ---
>
> > **C2.6 The article does not provide sufficient evidence of severe catastrophic forgetting in current large models.**
>
> As shown below, sequentially finetuning (Seq-FT) the large model (FrozenBiLM) without any continual learning technique leads to a 12.7% accuracy drop, which we believe is sufficient to show severe catastrophic forgetting. Thanks for the suggestion and we will add this baseline in Table 1 in the revision.
>
> |Method|iVQA|MSVD|MSR-VTT|LSMDC|ActivityNet|TGIF|Avg.|
> |-|-|-|-|-|-|-|-|
> |UpperBound|39.8|54.8|46.7|63.0|42.4|68.0|52.5|
> |Seq-FT|28.4|36.0|23.7|52.1|31.2|67.6|39.8|

---

> ### Author Response · Authors · 2023-11-19
> **Official Comment by Authors**
>
> > **C2.7 It is worth discussing whether there are unique challenges related to continual learning in the domain of VideoQA.**
>
> As mentioned on Page 2 (below Fig. 1) in the main draft, in contrast to unimodal tasks like image classification, VidQA involves both video and text inputs, necessitating the model to jointly reason on both modalities. Within VidQA domains, question types may exhibit significant similarity (e.g., when, how, where), while the corresponding answers can vary considerably. The juxtaposition of a shared input space (video+question) and a diverse output space adds complexity to continual VidQA. Consequently, router-based continual learning methods may face challenges in accurately predicting domain identities, leading to inferior answer prediction accuracies.
>
> Our proposed method, DAM, effectively mitigates the issue of inaccurate routers by dynamically merging domain adapters (refer to Sec. 4.4). We appreciate the suggestion and will incorporate this discussion in the revision.
>
> ---
>
> *If our response addresses your concerns, please consider increasing the scores. Also, feel free to ask follow-up questions during the rebuttal period.*

---

### Official Review · Reviewer_t6Nd · 2023-11-08

**Soundness:** 3 good
**Presentation:** 3 good
**Contribution:** 2 fair
**Rating:** 5
**Confidence:** 3

**Summary:**

This paper proposes the Dynamic Adapter Merging (DAM) for video question-answering under Domain-Incremental Learning scenario, which is a rehearsal-free approach. DAM leverages the fusion of parameters from multiple adapters to mitigate the interference of erroneous predictions, thereby enhancing the performance of the model.

**Strengths:**

The paper is well organized and the proposed method is verified through many experimental results.
The DAM is straightforward and easy to follow.

**Weaknesses:**

The paper provides a detailed elaboration to the framework of the model. However, the authors do not explicitly mention the loss function used during the training of adapters.

The contributions of the paper may be insufficient.  Although the Introduction section mentions four contributions, these contributions revolve primarily around one aspect, i.e. related to combining domain-specific adapter learning and model merging techniques.

The proposed method may lack innovation as the idea of model merging techniques in deep/machine learning is frequently used. The non-parametric router function is simply based on cosine similarity with no improvements. However, the application of such a concept to Continual Learning does introduce somewhat novelty.

**Questions:**

Can such ideas bring about desired performance improvements when extended to class-incremental learning and task-incremental learning scenarios? Can the author incorporate some results to demonstrate the generalizability of the idea in the context of continual learning?

How were the experimental results in the article obtained? Were multiple runs conducted to obtain an average, or was only a single experiment performed? I would like to know the stability of the proposed method.

---

> ### Author Response · Authors · 2023-11-19
> **Official Comment by Authors**
>
> We appreciate your insightful review and constructive suggestions. In response, we have diligently addressed all raised concerns, and we believe our revisions effectively resolve them. Should any of your questions or concerns persist, please feel free to communicate with us.
>
> ---
>
> > **C1.1 The loss function used during the training of adapters.**
>
> We didn’t apply additional loss functions to the adapters. Our loss function is the cross-entropy loss between the predicted tokens and ground-truth answer tokens, which is the same as the original VidQA model, i.e. FrozenBiLM in our paper.
>
> ---
>
> > **C1.2 The contributions of the paper may be insufficient, i.e. related to combining domain-specific adapter learning and model merging techniques.**
>
> While we acknowledge drawing inspiration from both continual learning and model merging, our work **introduces a novel technique**, "Dynamic Adapter Merging (DAM)," previously unexplored in both domains. Unlike existing methods that yield a single or a set of fixed models, DAM innovatively generates a **personalized expert model for each testing sample** with minimal overhead, which is not a simple adaption or combination of existing model merging methods and continual learning methods.
>
> Furthermore, our novelty extends to the **in-depth analyses** (as highlighted by [thbM] and [DkBm]) in Sec. 4.3 and Sec. 4.4, detailing how and when model merging can enhance the effectiveness of the router-based technique in the continual learning domain.
>
> Finally, the empirical results of DAM could be **inspiring to the model merging community**. As shown in the table below, we compare DAM with the other merging approaches, including average merging and RegMean (Jin et al., 2022), while all the approaches merge the same set of domain models that are individually finetuned on each dataset.  Unlike the other approaches, DAM determines the merging ratios for domain adapters based on the input instance, and this flexibility makes DAM outperform RegMean by **6.0%** and average merging by **7.5%** in average accuracy. The results show the potential of the proposed selective and dynamic merging strategy to inspire model-merging communities.
>
> |Method|iVQA|MSVD|MSR-VTT|LSMDC|ActivityNet|TGIF|Avg.|
> |-|-|-|-|-|-|-|-|
> |Avg. Merging|**38.0**|45.7|27.7|54.5|27.0|56.6|41.6|
> |RegMean|36.6|49.7|32.5|54.0|27.7|57.8|43.1|
> |DAM (Ours)|36.5|**51.6**|**39.5**|**63.0**|**36.5**|**67.7**|**49.1**|
>
> Thanks for your suggestion and we will highlight these contributions in our revised manuscript.
>
> ---
>
> > **C1.3 The non-parametric router function is simply based on cosine similarity with no improvements.**
>
> As demonstrated in Table 2, Section 4.3 of the main paper, our straightforward cosine-similarity-based router **surpasses all existing counterparts** with more complex router design, including MLP-based learnable routers proposed by L2P and CODA-Prompts, as well as the KMeans-based router employed in S-Prompts. Notably, our simple non-parametric router exhibits a **2.7%** higher accuracy in domain-identity prediction compared to the second-best router while having no training cost and faster inference speed. We also tried more complex router functions on the DAM framework. As shown in the table below, their performance is only comparable to our router. Thus, we keep our router simple but effective in the DAM approach.
>
> |Router|iVQA|MSVD|MSR-VTT|LSMDC|ActivityNet|TGIF|Avg.|
> |-|-|-|-|-|-|-|-|
> |GMM|38.5|55.1|43.4|63.0|31.2|65.4|49.4|
> |Learnable MLP|39.1|49.9|42.9|63.0|31.1|67.4|48.9|
> |Ours (cos.sim.)|39.1|53.6|42.2|63.0|36.3|66.8|50.2|
>
> ---
>
> > **C1.4 Generalizability: Can such ideas be extended to Class-Incremental Learning (CIL) and Task-Incremental Learning (TIL) scenarios?**
>
> We thank the reviewer for the valuable suggestion. We report both the CIL and TIL results in **G2 in General Response**. Results show that DAM is capable of outperforming the state-of-the-art continual learning approach under all settings.
>
> Additionally, In Sec 4.5 and Tab. 4 in the main paper, we show DAM can be applied to the visual (image)-QA task using a different large model BLIP2 on 4 ImageQA datasets.
>
> ---
>
> > **C1.5 Stability: How were the experimental results in the article obtained, multiple runs or a single run?**
>
> The results in our main table (Tab. 1) are obtained with 5 different random seeds. We report an average accuracy of **50.23 ± 0.12%**. Furthermore, in Tab. 9 (Ablations on domain orders) of the Appendix (following the reference section in the main paper PDF), we randomly sampled five different domain orders, observing consistent average accuracies (**50.56 ± 0.26%**).
>
> These findings affirm the stability of our method to variations in both random seeds and domain orders.
>
> ---
>
> *If our response addresses your concerns, please consider increasing the scores. Also, feel free to ask follow-up questions during the rebuttal period.*

---

### Author Response · Authors · 2023-11-19
**General Response to All Reviewers**

### General Response to All Reviewers

We thank all the reviewers for their insightful suggestions. We appreciate that reviewers agree that our paper is **well presented** [t6Nd, thbM, DkBm], the **first attempt** to address the issue of continual VidQA [t6Nd, thbM, DkBm]; the proposed method DAM is **easy to follow** [t6Nd, thbM, DkBm], **simple yet effective** [t6Nd, thbM, DkBm]; the ablation studies are **adequate** [t6Nd, yoGd], **comprehensive and in-depth** [thbM, DkBm].


The primary questions asked by the reviewers are centered around the **benchmark setup** and **generalizability**, which we have diligently addressed in the responses.

---

### **G1. Benchmark Setup**

> **Dataset Diversity [thbM]**

In our benchmark setup, we experimented on **6 datasets that come from multiple domains** (e.g. instructional videos, social videos, movies, long action videos, and short Tumblr GIFs) and are collected at different periods, which are diverse enough. We believe the confusion is caused by the question of **whether domains should be orthogonal**, which is related to the definition of domain that we will discuss shortly.

> **Definition of domains [DkBm]**

Our definition of the domain diverges from the conventional definition in the continual learning community. We define a **domain as a dataset collected at a specific period** (e.g. the datasets collected in 2011 and 2021 belong to different domains), taking into account of both 1) distribution-shifts over time and 2) domain (conventional) gaps.  This definition is motivated by the **practical considerations** outlined in our Introduction, and the need for the current commercial large models (like ChatGPT) to frequently update their knowledge to date. Therefore, our domains can be similar (e.g. MSVD-QA and MSRVTT-QA) and distinct (e.g. iVQA and ActivityNet-QA), which we believe is more in line with real-world applications.

---

### **G2. Generalizability**

We agree that the conventional definition (i.e. orthogonal domains) could lead to better model analysis. We showcase the **generalizability** of our method by further experimenting with Domain-Incremental Learning **(DIL) on more diverse domains**, as well as Class-Incremental Learning (**CIL**) and Task-Incremental Learning (**TIL**) scenarios.

> **DIL on more diverse domains** [thbM, DkBm]

We follow reviewers’ suggestion to further experiment on **more diverse VidQA domains**. This setup incorporates 4 originally benchmarked domains: social videos (MSVD), instructional videos (iVQA),  movie videos (LSMDC), and long activity videos (ActivityNet). Additionally, we introduce 3 new domains: indoor human videos (**AGQA**), traffic videos (**TrafficQA**), and virtual videos (**Env-QA**) as [thbM] suggested.

The table below shows DAM’s answer prediction accuracy and the router’s domain identity prediction accuracy. DAM only has **-1.1%** forgetting on this setting, which is even **1.2% less than on our original 6-domain setting**. This is because the router performs better on the domains (e.g. AGQA, EnvQA, TrafficQA) that are significantly different from the others. Our **DAM is better at dealing with more diverse domains** as they are easier to distinguish by the router function.

|Method|MSVD|iVQA|LSMDC|ActivityNet|AGQA|Env-QA|TrafficQA(½)|Avg.|
|-|-|-|-|-|-|-|-|-|
|Upper-Bound|56.6|39.7|63.0|42.2|63.4|32.3|67.8|52.1|
|DAM|54.0|39.3|63.0|37.4|63.3|32.0|67.8|51.0|
|Router Acc. of DAM|60.7|83.7|100|78.4|99.9|99.2|99.7|88.7|

> **Class-Incremental Learning (CIL) [t6Nd]**

We treat each unique answer as a class and adhere to a protocol commonly employed in continual image classification (Wang et al. 2022). Specifically, we experiment on two settings:1) **10 tasks split from MSRVTT-QA** with non-overlapping classes between tasks, and 2) **4-Datasets** (iVQA, MSVD, LSMDC, ActivityNet), excluding samples with overlapping answers across datasets. The results, presented below, indicate that DAM consistently outperforms S-Prompts, achieving **18.2%** and **8.5%** improvement on average accuracy on MSRVTT-QA 10-tasks and 4-Datasets respectively.

||MSRVTT-QA 10-tasks||4-Datasets||
|-|-|-|-|-|
|Method|Average Acc.|Forgetting|Average Acc.|Forgetting|
|Upper-Bound|47.3|-|51.6|-|
|S-Prompts|15.4|-23.5|42.2|-3.3|
|DAM (Ours)|**33.6**|**-13.7**|**50.7**|**-0.9**|


Note S-Prompts is compared with prompt-based multi-task finetuning when calculating the forgetting.

Wang, Zifeng, et al. "Learning to prompt for continual learning." CVPR 2022.

> **Task-Incremental Learning [t6Nd]**

We treat each dataset as a task. Unlike DIL or CIL, **TIL is provided with task indexes for inference** and thus DAM using task-specific adapters could overcome forgetting, i.e. DAM will always use the adapters of the task that the testing instance belongs to.

|Method|iVQA|MSVD|MSR-VTT|LSMDC|ActivityNet|TGIF|Avg.|
|-|-|-|-|-|-|-|-|
|Upper-Bound|39.7|56.6|46.7|62.9|42.2|67.8|52.6|
|DAM|39.8|54.8|46.7|63.0|42.4|68.0|52.5|

---

### Author Response · Authors · 2023-11-21
**Official Comment by Authors**

# A new revision is submitted

Dear reviewers, we have also submitted a revision based on your valuable feedback (all the updated contents are marked in brown). The changes are listed below:

1. On page 2: we clarified the **contribution paragraph** [t6Nd], emphasizing the novel concept of "**dynamic merging**".
2. On page 5: we clarified the definition of the symbol `k` [yoGd].
3. On page 6, Table 1: we added the Seq-FT (sequentially fine-tuned) baseline to show the **severe catastrophic forgetting in current large models** [thbM]; we changed the **upper bounds** from individually fine-tuned baselines to multi-task fine-tuned ones [DkBm].
4. On page 15, Appendix A (Implementation Details): we clarified the **loss function** [t6Nd], the definition of **pretrained model f** [DkBm, yoGd], the **network structure of adapters** [yoGd].
5. On page 15, Appendix B (**Efficiency analysis of Adapters**): We add the analysis of the size and computational cost of adapters for inference [DkBm].
6. On pages 16, 17, and 18, Appendix C, D, E, F: we also added **DAM as a model merging technique** to showcase the novelty [t6Nd], **generalizability studies** [t6Nd, thbM, DkBm], **design choices of router function** [t6Nd], **comparison with in-context learning** [thbM].

*Should any of your concerns persist, please feel free to ask follow-up questions. If our response addresses your concerns, please consider increasing the scores.*

---

### Author Response · Authors · 2023-11-21
**Gentle reminder of the rebuttal deadline**

Dear reviewers,

As the discussion period concludes tomorrow (**Nov 22, 2023**), we would like to inquire whether our response has effectively addressed your queries. Your reconsideration of scores is appreciated. For any additional questions, we are at your disposal before the deadline. Thank you for your time and contribution to the review process.

Warm Regards,
Authors

---

### Author Response · Authors · 2023-11-22
**New Results on Image Classification Task and Thanks to Reviewers**

# New Results on Image Classification Task

Addressing reviewers' concerns regarding generalizability, we further apply our method **DAM to the image classification task** on the Domain-Incremental Learning (DIL) scenario.
Due to time constraints, we only conducted experiments on the standard CORe50 benchmark dataset, comprising 50 categories across 11 domains. Following the literature, we continually train the model on 8 domains (120K images) and evaluate the remaining 3 domains (40K images). The pretrained backbone model is ViT-B/16, the same as the one used in prior work.

Our results reveal that DAM surpasses the current state-of-the-art, S-Prompts, by **9.32%**, underscoring its considerable potential to be applied to image classification tasks.

| Method                             | Average Acc      |
|------------------------------------|------------------|
| Upper-bound                        | 94.59 ± 0.21     |
| DyTox (w/ memory buffer for old domains) | 79.21 ± 0.10 * |
| LwF                                | 75.45 ± 0.40 *  |
| L2P                                | 78.33 ± 0.06 *  |
| S-Prompts                          | 83.13 ± 0.51 *  |
| DAM (Ours)                         | **92.45 ± 0.25**     |

\* indicates results are copied from (Wang et al. 2022).

---

# Many Thanks to Reviewers

In the rebuttal and our main paper, we systematically validated the generalizability of our proposed DAM method across diverse scenarios:

1. Domain-Incremental Learning (DIL) applied to **the image classification task** on the **ViT-B/16** model (above).
2. DIL extended to the **image-qa task** on the **BLIP2** model  (refer to Table 4 in the main paper).
3. DIL within **our original experimental setting** on the **FrozenBiLM** model (refer to Table 1 in the main paper).
4. **DIL across orthogonal domains** (refer to G2 in the General Response).
5. **Class-Incremental Learning (CIL)** assessed on two benchmarks (refer to G2 in the General Response).
6. **Task-Incremental Learning (TIL)** conducted on six tasks (refer to G2 in the General Response).
7. **Merging already-trained models** (refer to C1.2 in the rebuttal).

DAM consistently attains state-of-the-art (SOTA) results in all evaluated scenarios, spanning diverse settings, tasks, and models. Beyond the compelling empirical outcomes, we contend that our **thorough analyses** in Section 4.3 and Section 4.4 of the main paper contribute valuable insights, enhancing the community's understanding of our method and offering guidance for future research endeavors.

We express our sincere appreciation to all reviewers for **your valuable feedback, time, and efforts** dedicated to enhancing our paper. As the rebuttal deadline draws near, we regret the possibility of not having further in-depth discussions with you. We have done our best to address both your current and potential concerns. Wishing you a wonderful Thanksgiving day and a future filled with success and prosperity.

Best,
Authors

---

### Meta-Review · Area_Chair_QyEh · 2023-12-09

**Metareview:**

This paper tackles continual videoQA by training adapters specific to each domain (one dataset per domain).

The paper is borderline, having received mixed reviews (1 borderline accept and 3 borderline rejects).  Unfortunately, none of the reviewers engaged post-rebuttal, despite prompting by the AC.  The AC has carefully read through the reviews and author responses and especially appreciates the extensive experimentation added by the authors in an effort to address the reviewer concerns.

For the AC, the responses do not sufficiently address the concerns regarding
(1) the motivation or need for challenges on continual learning for VideoQA.  The explanation given in C2.7 lacks experimental or quantitative proof to substantiate.  Otherwise, this becomes simply a case of applying continual learning to VideoQA.
(2) lack of methodological contribution.  Most are borrowed from standard machine learning techniques.  As the authors themselves say, they simply adhere to existing literature in many of the design choices for the adapters.

As such, the AC recommends that the paper be rejected and resubmitted.

**Justification For Why Not Higher Score:**

Lack of sufficient exposition on the challenge or need for continual learning on VideoQA.

**Justification For Why Not Lower Score:**

N/A

---

### Decision · Program_Chairs · 2024-01-16

Reject